# Significance-based multi-scale method for network community detection and its application in disease-gene prediction

Ke Hu[1], Ju Xiang[2,3]*, Yun-Xia Yu📷[1]*, Liang Tang[3], Qin Xiang[3], Jian-Ming Li[3,4,5,6], Yong-Hong Tang[6], Yong-Jun Chen[6], Yan Zhang[2,3]

**1** School of Physics and Optoelectronic Engineering, Xiangtan University, Xiangtan, Hunan, People's Republic of China, **2** School of Computer Science and Engineering, Central South University, Changsha, Hunan, People's Republic of China, **3** School of Basic Medical Sciences, Changsha Medical University, Changsha, Hunan, People's Republic of China, **4** Department of Neurology, Xiang-ya Hospital, Central South University, Changsha, Hunan, People's Republic of China, **5** Department of Rehabilitation, Xiangya Boai Rehabilitation Hospital, Changsha, Hunan, People's Republic of China, **6** Department of Neurology, Nanhua Affiliated Hospital, University of South China, Hengyang, Hunan, People's Republic of China

* xiang.ju@foxmail.com, xiangju@csu.edu.cn(JX); yunxiayu@xtu.edu.cn(YXY)

**Data Availability Statement:** All relevant data are within the manuscript.

**Funding:** The following grants provided support for this study: National Natural Science Foundation of

## Abstract

Community detection in complex networks is an important issue in network science. Several statistical measures have been proposed and widely applied to detecting the communities in various complex networks. However, due to the lack of flexibility resolution, some of them have to encounter the resolution limit and thus are not compatible with multi-scale structures of complex networks. In this paper, we investigated a statistical measure of interest for community detection, *Significance* [Sci. Rep. 3 (2013) 2930], and analyzed its critical behaviors based on the theoretical derivation of critical number of communities and the phase diagram in community-partition transition. It was revealed that *Significance* exhibits far higher resolution than the traditional *Modularity* when the intra- and inter-link densities of communities are obviously different. Following the critical analysis, we developed a multi-resolution version of *Significance* for identifying communities in the multi-scale networks. Experimental tests in several typical networks have been performed and confirmed that the generalized *Significance* can be competent for the multi-scale communities detection. Moreover, it can effectively relax the first- and second-type resolution limits. Finally, we displayed an important potential application of the multi-scale *Significance* in computational biology: disease-gene identification, showing that extracting information from the perspective of multi-scale module mining is helpful for disease gene prediction.

## 1 Introduction

Complex systems, including the artificial and natural ones in the real world, can be properly described as complex networks that consist of vertices and links. Typical examples contain the social, biological, and computer information networks. Currently, it has been recognized that

China (Grant No. 61702054(JX), 81873780(JML));
Hunan Provincial Natural Science Foundation of
China (Grant No. 2018JJ3568(JX) and
2019JJ50697(LT)); Training Program for Excellent
Young Innovators of Changsha (Grant No.
kq1905045(JX) and kq1905047(LT)); Scientific
Research Fund of Education Department of Hunan
Province (Grant No. 17A024(JX),18B539(YZ),
19B072(QX)); Hunan Key Laboratory Cultivation
Base of the Research and Development of Novel
Pharmaceutical Preparations (No. 2016TP1029)
(JML); Hunan Provincial Innovation Platform and
Talents Program (No. 2018RS3105)(JML); Hunan
Provincial Science and Technology Department
and Human Provincial Health and Family Planning
Commission (Grant No.[2018]85)(YHT); Key
Project of Hunan Provincial Commission of Health
and Family Planning (Grant No.[2017]144)(YHT);
Hunan Provincial Science and Technology
Department Clinical Medical Technology Innovation
GuideProject(2018SK50711)(YJC).

**Competing interests:** The authors have declared
that no competing interests exist.

networked description of complex systems is a kind of useful approach to study the structures of and dynamical processes on these systems. Although networked structures are abstracted out from different complex systems, they exhibit many common topological properties [1, 2], such as small-world property, scale-free features and modularity. Among them, the modularity indicates that the networks are generally organized by the communities with dense inner-connections and sparse outer-connections. Community structure is a special perspective for understanding the structures and functions of complex networks and can also significantly affect the dynamical behaviors on networks [3–8]. For example, when the epidemic spreads on a network with clear community structure, the local targeted immunization will be more effective than global targeted one for preventing the epidemic outbreak [9]. Also, the cooperation under strong selection is closely related to the density of communities in social networks [10]. For another example, the genes associated with the same or similar disease phenotypes are usually involved in the same molecular module or pathways leading to the disease. Thus, community or module can be helpful for identifying the disease genes [11] and understanding the disease processes [12].

Benefited from the underlying implications, many community-detection methods have been proposed and developed during the past decade [13–30]. Typical examples include the spectral analysis [22], random walk [23–25], label propagation [30], dynamic evolutionary [26–29], and modularity optimization [31, 32]. They could effectively identify community structures in complex networks and are helpful for understanding their underlying functions. However, some of them were found to have respective scopes of application. For instance, phase transitions from detectable to undetectable community structures were found to exist in the methods based on modularity optimization and *Bayesian* inference [33–35]. In addition, some limitations have been uncovered for the recently proposed modularity density maximisation algorithm [36]. Especially, as a paradigm of some community-detection methods, *Modularity* was found to have the (first-type) resolution limit where communities below certain scale cannot be identified [37]. Similarly, the resolution limit has be also suffered by some other global measures. In order to solve or alleviate these problems, several schemes [15, 38, 39], such as the random walk network preprocessing [39] and the analysis of correlation between communities [15], have been recently suggested. Also, due to the important implication for biomedical research, the Disease Module Identification DREAM Challenge [40] have been initiated as a joint effort to comprehensively evaluate module identification methods on gene and protein networks. All these facts suggest that it is still necessary to investigate the community-detection methods in detail, aiming to understand the methods themselves and improve them. In particular, the resolution limit implies that community structures generally distribute at multiple scales [1], and thus it is naturally required to develop methods with a adjustable resolution to identify the multi-scale communities in complex networks.

In literatures, some methods with flexible resolution have been proposed to analyze multi-scale communities in networks based on distinct approaches [14, 41–45], such as the correlation between dynamics and multi-scale structures [25, 46–48], the local optimization of fitness functions [49], and Potts model [30, 44, 47, 50, 51]. However, a simplest and effective scheme for solving the problem of resolution limit is to introduce a tunable resolution parameter into the quantitative measure evaluating accuracy or quality of network partition [41, 45]. Particularly, one standard framework for the multi-resolution *Modularity* was proposed recently by using the general rescaling strategy [14] where several important measures have been well unified [41, 44, 45, 51].

It is well known that one of most popular methods is to detect the community structures by directly optimizing statistical measures (e.g., *Modularity* [52], *Hamiltonian* [44], Partition density [53, 54]). In reference [55], Traag et al proposed a statistical measure of interest for

community detection, called *Significance*. It evaluates how likely dense communities appear in random networks by,

$$S = \sum_s \binom{n_s}{2} D(p_s \parallel p)$$

$$= \sum_s \frac{n_s(n_s-1)}{2} \left[ p_s \ln \frac{p_s}{p} + (1-p_s) \ln \frac{1-p_s}{1-p} \right], (1)$$

where $n_s$ denotes the number of vertices in community $s$; $p_s$ is the link density of community $s$, i.e., the ratio of the number of existing links to the maximum within the community; $p$ denotes the link density of whole network, i.e., the ratio of the number of existing links to the maximum in the whole network; the sum runs over all communities. This measure was initially used to evaluate the significant scale of community structures, but it could also be directly optimized as a target function to search for the optimal community partitions [55]. It was shown that *Significance* has a good performance in some networks, due to its high resolution. But it is still a kind of single-scale method with limited resolutions and thus not compatible with the multi-scale structure in complex networks.

In this paper, we firstly discuss the critical behavior of *Significance* and analyze its resolution, by analytically deriving the critical number of communities in community-partition transition. Following the theoretical analysis, the multi-resolution method based on *Significance*, i.e., an extension of *Significance* to the multi-scale networks, is then designed by using a resolution parameter to adjust the random model. Finally, the multi-resolution *Significance* are tested experimentally on various baseline networks, to demonstrate the efficiency for identifying the multi-scale communities and resolving the problem of resolution limit.

## 2 Phase transition of *Significance* in community detection

In order to learn about the critical behavior of *Significance* in community-partition transition and provide a theoretical basis for designing the multi-resolution method based on *Significance*, we here conduct the critical analysis of *Significance* and discuss its resolution in community detection by analytically deriving the critical number of communities for community merging.

### 2.1 Critical analysis of *Significance*

For the sake of convenience of analysis, a set of computer-generated, called community-loop networks, are firstly introduced. In each community-loop network, a total of $r$ communities is connected one by one and each community has $n_c$ vertices. The probability of linking vertices within community and between two adjacent communities are denoted by $p_i$ and $p_o$, respectively. An example for the community-loop network is shown in Fig 1. To investigate the critical behavior of *Significance* in partition transition, we may assume a set of partitions where each partition contain $r/x$ groups of vertices and each group contains $x$ adjacent communities. For any partition of this kind, *Significance* can be then written as

$$\begin{aligned} S_x &= \frac{r}{x} \binom{x \cdot n_c}{2} D(p_x \parallel p) \\ &\approx \frac{r \cdot x \cdot n_c^2}{2} \left[ p_x \ln \frac{p_x}{p} + (1-p_x) \ln(1-px) \right], \end{aligned} \quad (2)$$

where $p_x = \frac{p_i}{x} + \frac{2(x-1)p_o}{x^2}$, $p = \frac{p_i + 2p_o}{r}$, and $1 - p \approx 1$ for large $r$-value. In the networks, the

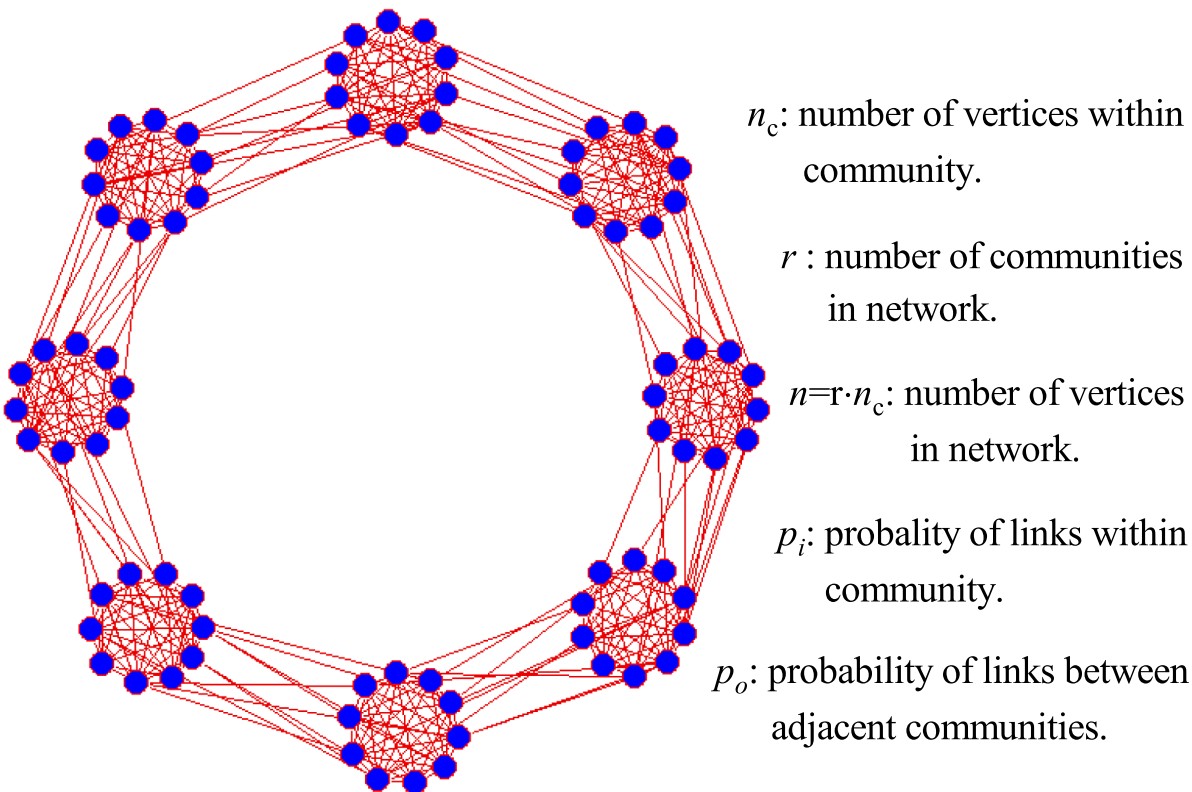

$n_c$: number of vertices within community.

$r$ : number of communities in network.

$n=r\cdot n_c$: number of vertices in network.

$p_i$: probality of links within community.

$p_o$: probability of links between adjacent communities.

**Fig 1. An example of community-loop network.** An example of community-loop network and the definitions of network parameters.

predefined community partition corresponds to the case with $x = 1$, while $x \geq 2$ means the partitions with communities merging.

*Significance* is a multivariate function, which is closely related to various network parameters. For the sake of visual illustration, Fig 2(a)–2(c) shows the curves of *Significance* with several different network parameters. As can be seen from Fig 2(a), for small $r$ values, $S(x)/S(1)$ monotonously decreases with the increase of $x$, so that $S(x)/S(1)$ is always less than 1 which indicates that any coalescence of communities do not occurs. But, for large $r$ values, a significant peak appears at $x = 2$ where $S(x)/S(1)>1$. This implies that some communities have merged into a single and large one. The similar situation can be also found from the curves of $S(r)$ with $x = 2$ and 3. From Fig 2(b), one can noted that all communities can survive separately and do not merge with each other when $r$ is relatively small. However, with the increase of $r$, $S(r, x)/S(r, 1)$ will increase continuously and be significantly larger than 1, which implies that communities have began to merge. Moreover, with the increase of $r$, $S(r, x = 2$ and 3) will be larger than others in turn, which indicates that the merging of communities for $x = 2$ and 3 will be preferred, compared with $x = 1$. In addition, with the increase of $p_o/p_i$, the *Significance* normalized by the number $m$ of links existing in network, i.e. $S/m$, decreases for all different $x$ values, and $S(x = 1, 2$ or 3) will be larger than others in turn (see Fig 2(c)). This means that the partition for $x = 1, 2$ and 3 will be preferred in turn. Other statistical measures, such as *Surprise* and *Modularity*, have similar phenomena, but different statistical measures have different critical points in partition transition.

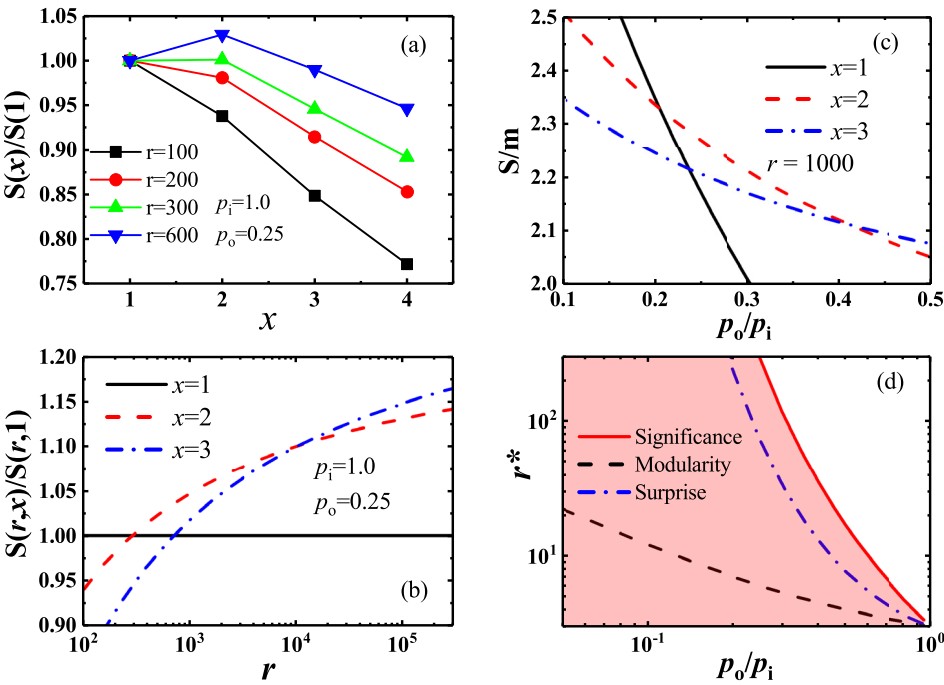

**Fig 2. Critical behavior of *Significance* in partition transition.** (a) Relations between *Significance* $S$ and the number $x$ of communities merging in the networks with different sizes. *Significance* is normalized by $S(x=1)$, i.e. the *Significance* for the pre-defined partition. (b) Relations between the normalized *Significance* for various $x$ and the number $r$ of pre-defined communities. (c) *Significance* as a function of $p_o/p_i$ for distinct $x$, where *Significance* is normalized by the number $m$ of links in the networks. (d) The critical number $r^*$ of communities for communities merging as a function of $p_o/p_i$, which represents the phase transition in network partition by three different methods, i.e., *Significance*, *Surprise* and *Modularity*.

## 2.2 Resolution of *Significance* in community detection

As we show above, the merging of communities may appear, or say, be allowed in some cases. When $S_x > S_1$, the partition identified by *Significance* will be the partition with communities merging while not the predefined one. In order to theoretically analyze the critical points of *Significance* for community merging, we focus on the transition of partition from $x = 1$ to $2$.

According to Eq 2, the critical condition reads $\binom{2n_c}{2}D(p_2 \parallel p) - 2\binom{n_c}{2}D(p_1 \parallel p) = 0$ (see Appendix A). By solving the equation for $r$, one can obtain the critical number of communities,

$$
\begin{aligned}
r^* &\approx \exp\Big\{\frac{1}{2p_2 - p_1}\big[p_1\ln(p_1) + (1-p_1)\ln(1-p_1) - 2p_2\ln(p_2) \\
&\quad - 2(1-p_2)\ln(1-p_2)\big] + \ln(p_i + 2p_o)\Big\} \\
&= (p_i + 2p_o)\exp\left[\frac{2H(p_2) - H(p_1)}{2p_2 - p_1}\right] \\
&= (p_i + 2p_o)\exp\left[\frac{1 + 2\Delta H/H(p_1)}{1 + 2\Delta p/p_1} \cdot \frac{H(p_1)}{p_1}\right],
\end{aligned}
\tag{3}
$$

where $p_2 = (p_i + p_o)/2$ $(p_1 = p_i)$, $H(y) = -y\ln(y) - (1-y)\ln(1-y)$ is the information entropy; $\Delta H = H(p_2) - H(p_1)$ and $\Delta p = p_2 - p_1$. Obviously, the critical number of communities at partition transition point is extremely dependent on the variations of information entropy resulted

from the link-density changes in a community. For comparison, the critical number of communities for *Modularity* is also derived by $r^* = p_i/p_o + 2$. Because of the complexity of nonlinearity of *Surprise*, its critical point is difficult to be obtained analytically.

In Fig 2(d), we presented the phase diagrams of partition transition for *Surprise*, *Modularity* and *Significance* according to their critical numbers, where the partition with community-merging appears above the corresponding curves, while not below the curves. It is not unexpected that the resolution of *Significance* decreases with the increase of $p_o/p_i$. Because the number of links between communities gradually increases and the differences between the intra- and inter-link densities will decrease with the increase of $p_o/p_i$, the community structures become more and more unclear. Naturally, the resolution of *Significance* decreases. Similar situations can be found for *Surprise* and *Modularity*. However, it should be noted that *Significance* has highest resolution, while *Modularity* has lowest resolution. Especially for small $p_o/p_i$ values, the critical number $r^*$ of *Significance* dramatically increases with the decrease of $p_o/p_i$, so that it is far larger than that of *Modularity*. This implies that *Significance* generally tends to split communities in networks, especially for the networks with low inter-community link density, and can usually detect more communities than other methods.

## 2.3 Experimental results of resolution limit

In this section, we experimentally tested the resolutions of these measures based on a direct optimization. Usually, one can employ the Normalized Mutual Information (NMI) [56], a measure of similarity originating from information theory, to estimate the performance of the community-detection methods. In fact, NMI presents the similarity between two community partitions, and reveal the amount of community information correctly obtained in the networks with known community structures. If two community partitions are matched perfectly, NMI will be equal to 1. While the smaller NMI means less matching. Also, the Fraction (Fr) of vertices affected by merging of communities is calculated to demonstrate the performance of these measures. As can be seen from Fig 2, it will be not easy to identify the predefined communities when $p_o/p_i$ is very large, i.e., when the difference between the inter- and intra-community link densities is very small. In fact, some communities have merged into one group at this time, which indicates the emergence of the resolution-limit problem. As a result, these methods can not effectively identify all predefined communities in the network, so that their NMIs decrease with the increase of $p_o/p_i$ (Fig 3(a) and 3(c)). Clearly, it is also seen from the variations of Fr which are depicted in Fig 3(b) and 3(d). Moreover, the more the predefined communities, or the larger the network size, the quicker the communities merge for these methods. However, with the increase of $p_o/p_i$, NMI of *Significance* is larger and the significant reduction is later than that of *Modularity*, which indicates that *Significance* can remarkably outperform *Modularity*. In addition, by combining Figs 2(d) with 3, one can found that *Significance* has a higher resolution than *Surprise*, but NMI of *Significance* seemly decreases more quickly than that of *Surprise*. This is because *Surprise* has the so-called "potential well" effect. General greedy optimization algorithms are difficult to get across the "potential well" to find the final optimal solution [57]. However, *Significance* doesn't encounter the "potential well" effect (see Appendix B).

Further, we apply the measures to a set of networks with tunable sizes, i.e., Lancichinetti-Fortunato-Rachicchi (LFR) networks [58]. They have heterogeneous structures and some other statistical properties exhibited by many real-word networks. In addition, the NMI may result significantly non-zero when two random partitions with large numbers of groups are compared, because random coincidences become likely in this case. Similarly, it may result in artificially large values, even when two non-random partitions are compared if these have a

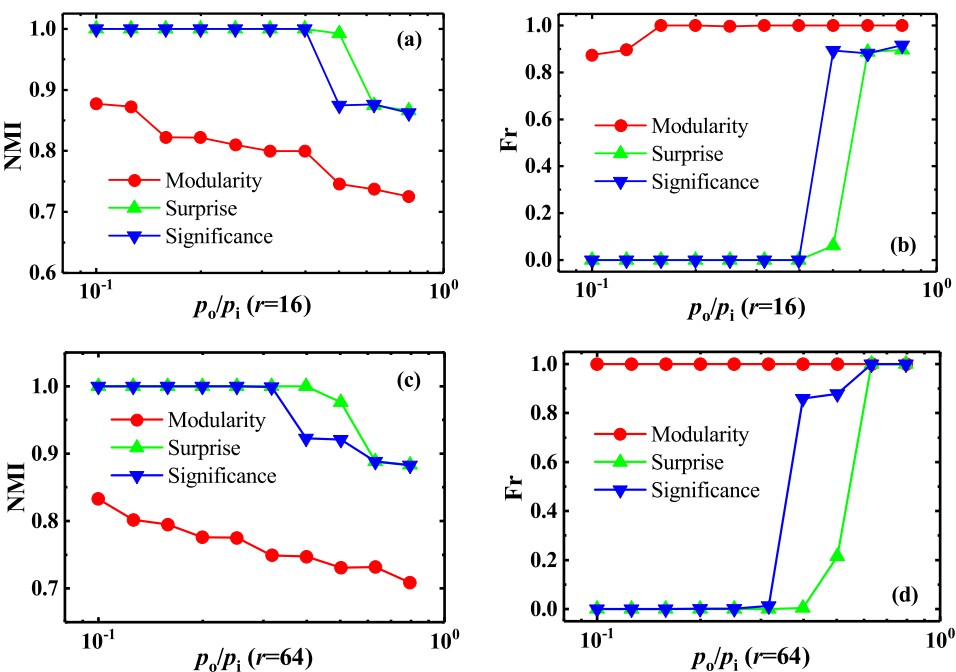

**Fig 3. Normalized mutual information and fraction of vertices affected by merging of communities in community-loop networks.** Normalized mutual information (NMI) calculated from three different methods, as a function of $p_o/p_i$ in the community-loop networks with (a) $r = 16$ and (c) $r = 64$, respectively. Fr as a function of $p_o/p_i$ in the networks with (a) $r = 16$ and (c) $r = 64$.

large number of groups. To counter balance for such bias, several metrics alternative to the NMI were introduced [59–61]. It seems that Significance tends to favor the detection of small-scale structures, potentially returning partitions with more communities (i.e. groups) than other methods such as those based on Modularity Maximization. It is convenient, then, to use one of these alternative metrics to judge the benefits of the Significance as compared to that of Modularity. Therefore, we also employed two other metrics: the adjusted mutual information (AMI) [59] and the adjusted Rand index (ARI) [59], to comprehensively estimate the performances of these community-detection methods. As is shown in Fig 4(a)–4(c), all three metrics indicate that *Significance* gets a somewhat better performance than *Surprise*, and significantly overcomes *Modularity*, especially for the LFR networks with a large mixing parameter. Of course, with the increase of the mixing parameter, it will be also difficult that both *Significance* and *Surprise* identify all predefined communities, because the resolution-limit problem will appear and even become more severe. Finally, the network-size effects is also investigated to assess their performances, which are shown in Fig 4(d)–4(i). Interestingly, with the increase of network size, NMI, AMI and ARI for both Significance and Surprise gradually increase, while decrease for Modularity, indicating that Significance and Surprise have better performance for the large networks than Modularity.

## 3 Analysis of multi-resolution method based on *Significance*

### 3.1 Multi-resolution *Significance* and its resolution

In view of the emergence of multi-scale community structures in many realistic networks, both *Modularity* and *Surprise* have been extended to the corresponding multi-scale versions. However, the traditional *Significance* is still a single-scale method in spite of its high resolution

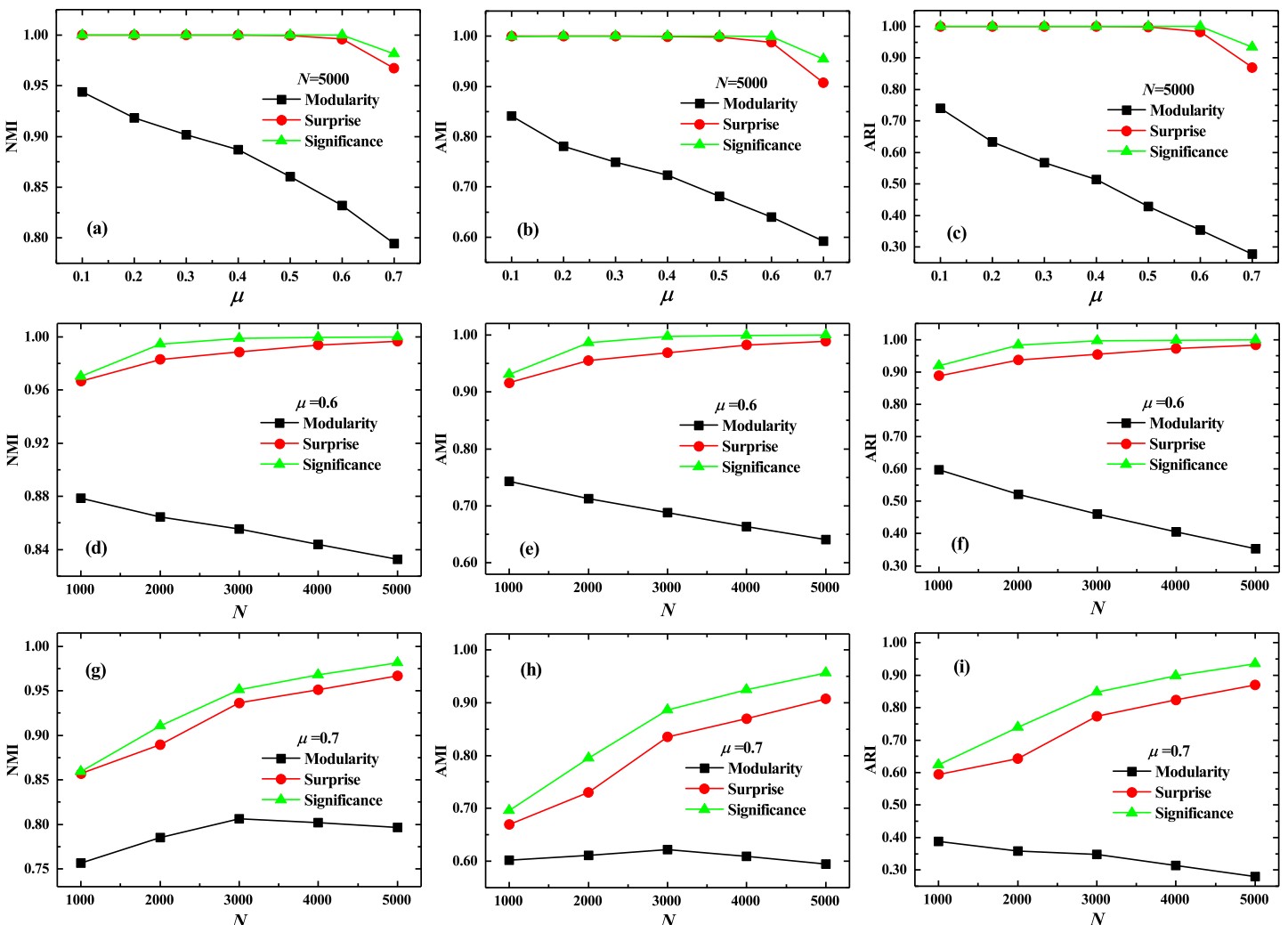

**Fig 4. Normalized mutual information, adjusted mutual information and adjusted Rand index in LFR networks.** Three different metrics: (a) Normalized mutual information (NMI), (b) Adjusted mutual information (AMI) and (c) Adjusted Rand index (ARI), obtained by three different methods versus the mixing parameter in the LFR networks [58], where the mixing parameter in the LFR networks is defined as a free parameter to adjust the ratio between the external degree of each vertex with respect to its community and the total degree of the vertex. (d)-(i) The network-size effects for these metrics in LFR network with two typical mixing parameters.

and well performance for community detection. As we know, in order to obtain the corresponding multi-scale methods, a simplest and effective approach is to introduce a additional variable to adjust the weight of the referential random network (i.e. the null model). Because *Significance* depends on the difference between the link density of community and the link density of network, i.e., the expected value in a null model, we can easily extend the traditional *Significance* to a multi-scale version by introducing a resolution parameter to adjust the link density of network. So, that is to say, we may modify the traditional *Significance* as

$$
\begin{aligned}
S(\gamma) &= \sum_s \binom{n_s}{2} D(p_s \parallel \tilde{p}) \\
&= \sum_s \frac{n_s(n_s-1)}{2} \left[ p_s \ln p_s \tilde{p} + (1-p_s) \ln \frac{1-p_s}{1-\tilde{p}} \right],
\end{aligned}
\tag{4}
$$

where $\tilde{p} = \gamma \cdot p$ and $\gamma$ is the resolution parameter. Of course, the critical point of *Significance*

for communities merging should be also changed accordingly as

$$r^* \approx \gamma(p_i + 2p_o)\exp\left[\frac{2H(p_2) - H(p_1)}{2p_2 - p_1}\right].$$  (5)

For comparison, we also give the critical point for the multi-resolution *Modularity*, i.e., $r^* = \gamma(p_i + 2p_o)/p_o$. Obviously, with the increase of the resolution parameter, the resolution of these methods increases.

The multi-resolution *Modularity* can identify the communities that are undetectable for the original *Modularity* and the community structures at different scales, by varying the resolution parameter. Similarly, the multi-resolution *Significance* is also able to detect the communities beyond the resolution of original *Significance* and identify the community structures at different scales. Moreover, one can use any effective algorithms to search for the optimal values of the statistical measures for community detection. Here, we use the Louvain procedure. Louvain process is a widely used and efficient algorithm, though its exact computational complexity is not known. Most of its computational effort is spent on the optimization at the first level, taking a time $O(nk_m f)$ if we control the maximal iteration times, where $n$ is the number of nodes, $k_m$ is the mean degree of nodes, and $f$ is the number of operations of calculating $S$-value each time (on average the number of communities that each node connects to is less than the number of neighbors of the vertex).

## 3.2 Attack to the first-type resolution limit

As discussed above, because of the lack of flexible resolution, the original *Significance* and *Modularity* cannot identify communities below a certain scale, which will be merged into large communities. This means the appearance of the first-type resolution limit. In order to test the performance of these multi-resolution methods, we apply them to detect the multi-scale communities in both the community-loop network and LFR one. As shown in Figs 5 and 6, the multi-resolution *Significance* and *Modularity* can successfully solve the resolution-limit problem, where those predefined communities have been identified correctly at a suitable resolution. Of course, as should be pointed out, with the increase of $p_o/p_i$ values, the needed value of the resolution parameter for solving the first-type resolution limit increases. This means that it is more and more difficult to identify the predefined communities.

## 3.3 Strong tolerance to the second-type resolution limit

When applying a multi-resolution method to detect the multi-scale communities in a network, some unstable and large communities can be split before small communities become detectable. This phenomenon, called as the second-type resolution limit, can be generally encountered only if the difference of community size is large enough [43, 45]. Therefore, in order to comprehensively assess the community detection methods, the second-type resolution limit should be also considered as a important criteria for assessing their effectiveness. Here, a set of Fortunato and Barthélemy (FB) networks [37] was adopted to test the ability of the proposed multi-resolution *Significance* against the second-type resolution limit. For comparison, the multi-resolution *Modularity* have been also tested experimentally.

As is demonstrated in Fig 7(a), when the size of the large community is not significantly larger than that of the small community, the multi-resolution *Modularity* may detect all predefined communities in the FB network by choosing a suitable resolution parameter, whose partition was marked by $N_d = 4$ in Fig 7(a). In fact, due to the small difference of size among these predefined communities in the FB network, *Modularity* does not suffer from the second-type resolution limit at all. However, when the relatively large difference of community sizes is set

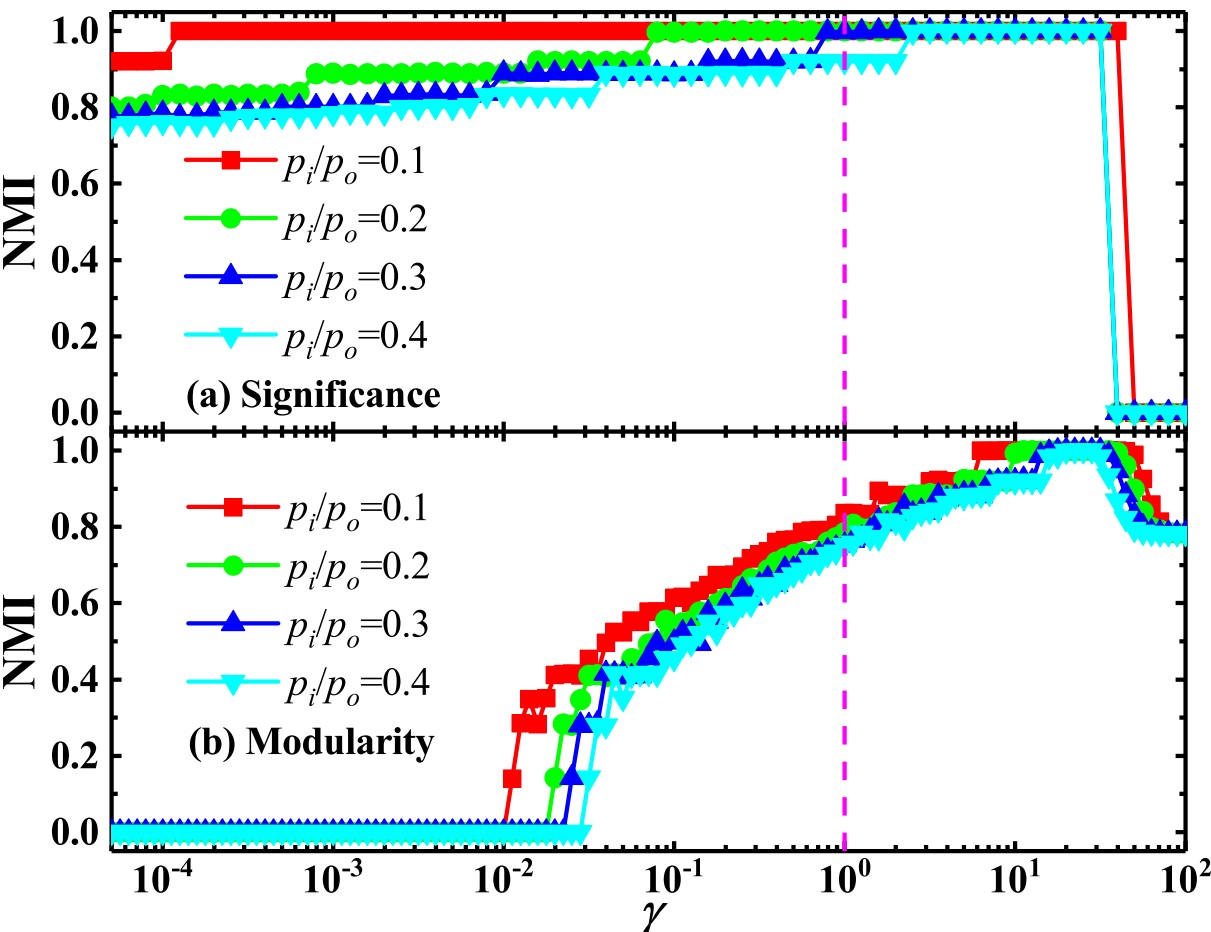

**Fig 5. Normalized mutual information in community-loop networks with different $p_o/p_i$ values.** NMI calculated by using (a) *Significance* and (b) *Modularity*, as a function of resolution parameter $\gamma$, in the community-loop networks with $r = 64$ and several typical values of $p_o/p_i$.

in the FB network, it become difficult to detect the predefined partition. As shown in Fig 7(b), in order to detect the predefined and small communities, one has to increase the resolution parameter, but at this time, the large communities have been split into many small cliques by *Modularity*. Therefore, the multi-resolution *Modularity* can not solve, at least not well alleviate the problem of the second-type resolution limit, in spite of its adjustable resolution parameter.

Interestingly, the multi-resolution version of *Significance* is able to correctly identify the predefined community partition in the two networks with small and large community-size difference, as shown in Fig 7(c) and 7(d). This means the strong tolerance of the multi-resolution *Significance* for the second-type resolution limit. It can detect the community structures in the networks better than *Modularity*.

### 3.4 Effectiveness of identifying multi-scale community structures

The flexible resolution of multi-resolution *Significance* can help in detecting communities at multiple scales. To further test the ability of multi-resolution *Significance* to identify multi-scale community structures, we firstly employed two kinds of computer-generated networks with a well-defined hierarchical community structure, i.e., the homogeneous hierarchical network [62] and the heterogeneous hierarchical network [63]. They have been widely used as a

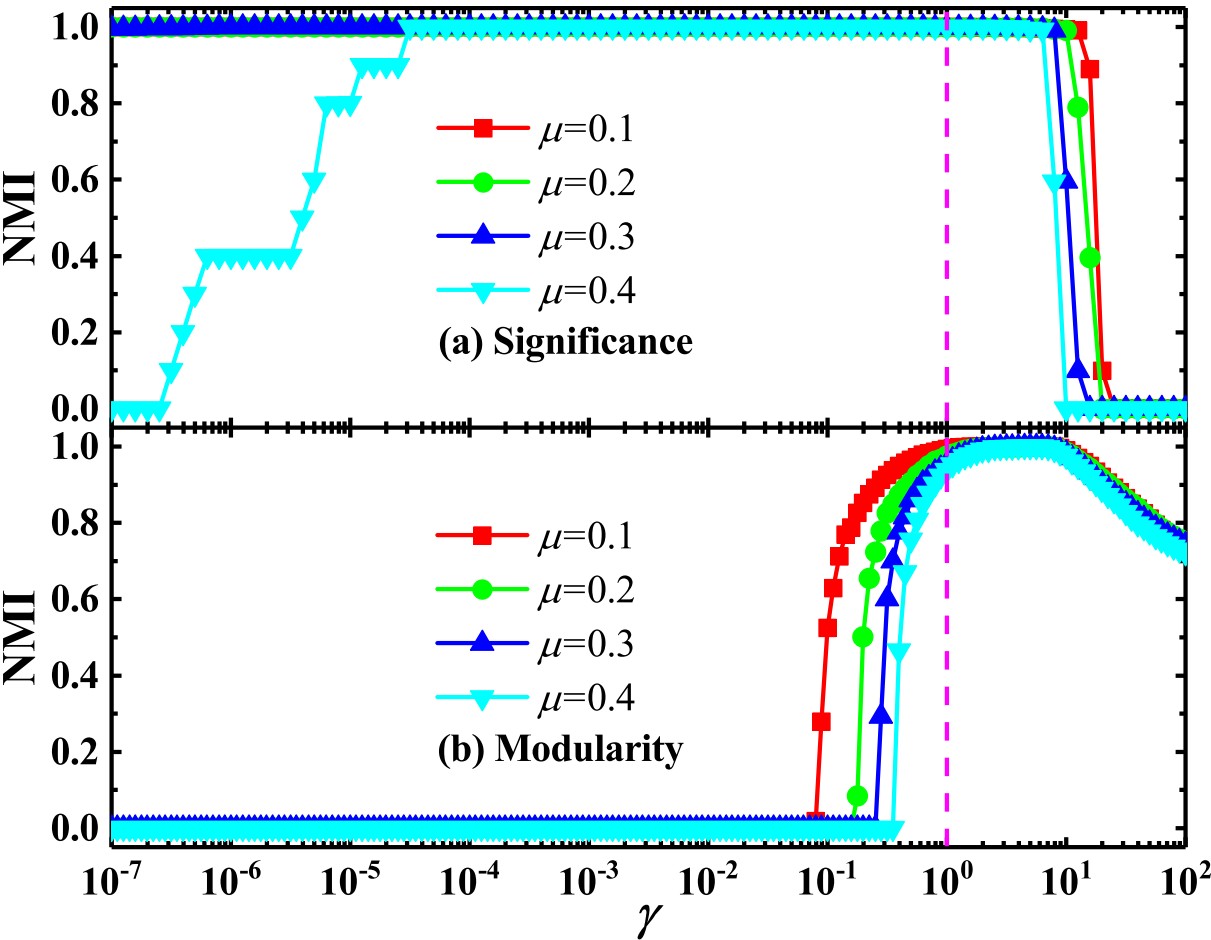

**Fig 6. Normalized mutual information in LFR networks with different _μ_ values.** NMI calculated by using (a) *Significance* and (b) *Modularity*, as a function of resolution parameter _γ_, in the LFR networks with _N_ = 1000 and several values of _μ_. Other parameters are the same as in Fig 4.

benchmark for testing various community-detection methods. The homogeneous hierarchical network [62] is generally constructed to include two hierarchical levels of communities. Here, we assume that each homogeneous hierarchical network is composed of 256 vertices. At the first level (L1), these 256 vertices are equally divided into 16 groups, and thus every group includes 16 vertices. The second level L2 contains four relatively large groups where each of them is composed of four different groups of the above first level. The vertices in the network are connected by setting the number of internal links of each vertex within the first-level community and within the second-level one as $k_{in0}$ and $k_{in1}$, respectively. The number of links with any other vertex at random in the network is 1. The heterogeneous hierarchical networks we generate are as follows: 1000 vertices and two predefined hierarchical levels are firstly prescribed. The number of links is controlled by fixing the average degree of vertices to be 20. The maximum degree is set to be 50. The sizes of these micro communities distributed between 10 and 25 which constitute the micro level of a homogeneous hierarchical network, while the sizes of communities at the macro level are changed from 50 to 100. In addition, two mixing parameters, $\mu_1$ and $\mu_2$, are treated as free ones to adjust the fraction of links between vertices belonging to different macro communities and belonging to the same macro but not micro

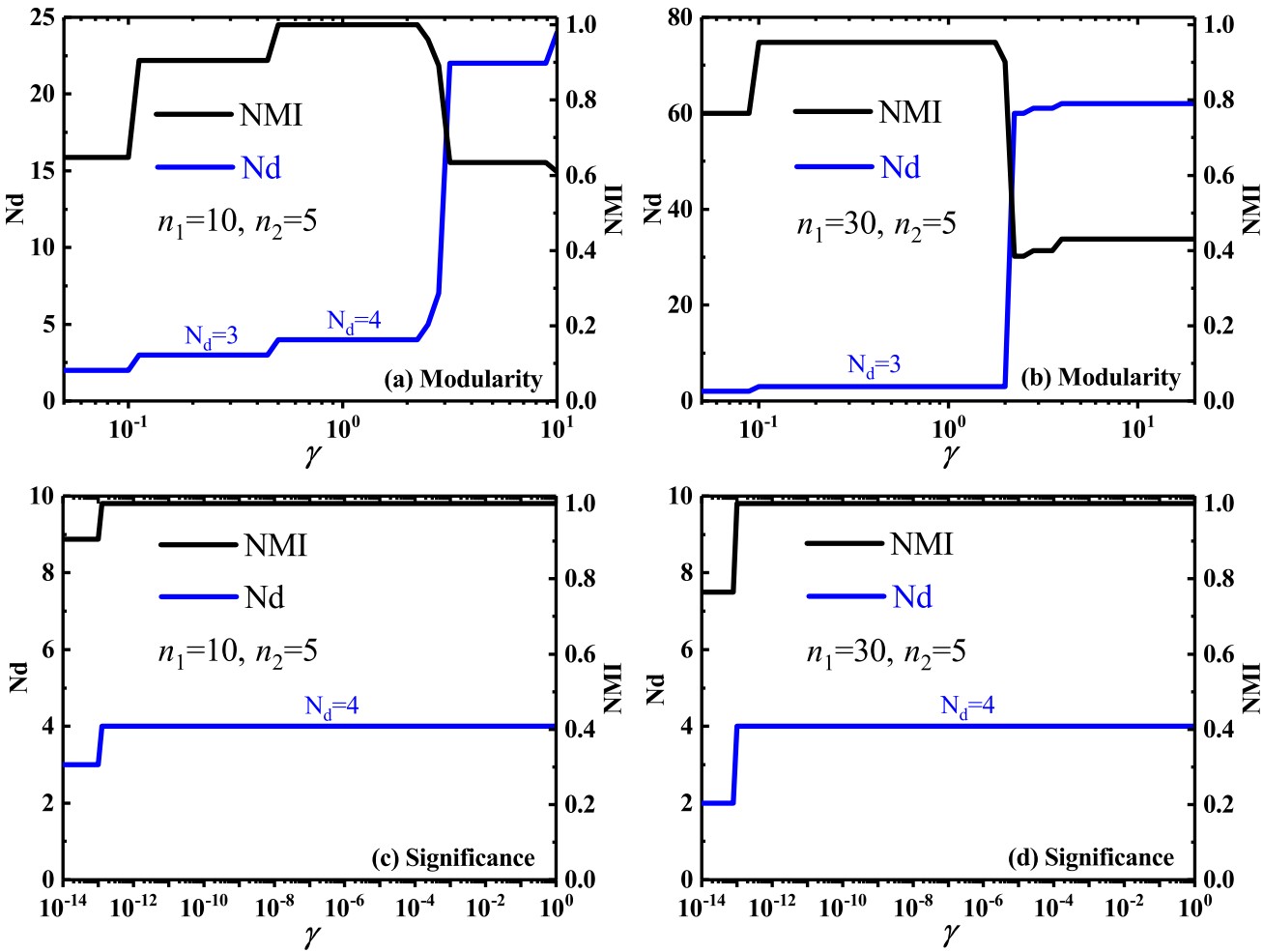

**Fig 7. Number of identified communities and normalized mutual information in FB networks.** Number of identified communities Nd and Normalized mutual information NMI, by multi-resolution *Modularity* (a)-(b) and multi-resolution *Significance* (c)-(d), versus the resolution parameter $\gamma$, in the FB networks. Each FB network is composed of four predefined communities, i.e., two large communities with $n_1$ vertices and two small communities with $n_2$ vertices. The partition corresponding to Nd = 3 indicates that two predefined small cliques have been identified as a single community, while the situation for Nd = 4 means that all predefined four communities have been exactly identified.

community, respectively. The remaining parameters are adopted according to the default values of program.

In the homogeneous hierarchical network, all communities predefined at two different levels can be well detected by both the multi-resolution *Modularity* and *Significance*. In Fig 8, the network partitions predefined at two levels have been correctly identified which are marked by L1 and L2, respectively. When comparing Fig 8(a) and 8(c) with Fig 8(b) and 8(d), one can find that the leap of Nd from L1 to L2 will occurs at a smaller value of the resolution parameter, which means that the smaller the number of internal links of each vertices at the second level ($k_{in1}$), the smaller the required resolution parameter can be for detecting the communities at the second level L1. Because $k_{in1}$ controls the density of links between the first-level communities, the small $k_{in1}$ means the sparse links among the first-level communities which leads to the first-level communities are easily identified.

In Fig 9, we find that the original *Modularity*, i.e., the multi-resolution *Modularity* with $\gamma = 1$, could only identify these heterogeneous communities predefined at the macro level, while

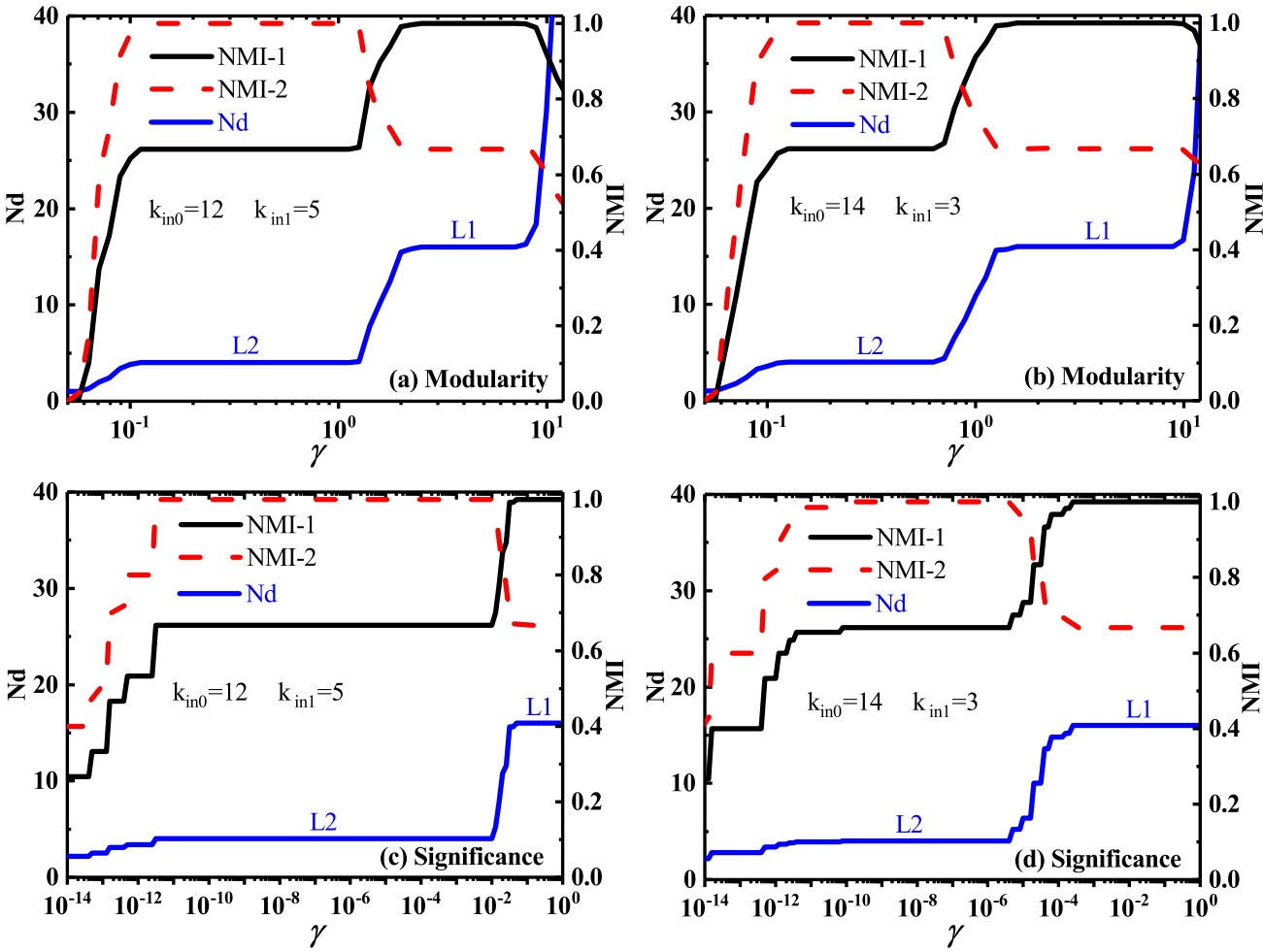

**Fig 8. Number of identified communities and normalized mutual information in homogeneous hierarchical networks.** Number of identified communities Nd and Normalized mutual information NMI, by the multi-resolution *Modularity* (a)-(b) and the multi-resolution *Significance* (c)-(d), as a function of resolution parameter $\gamma$, in the homogeneous hierarchical networks with two hierarchical levels of homogeneous communities. L1 and L2 are marked to highlight the two predefined scales in the networks. NMI-1 denotes the NMI between the identified partition and the predefined partition at the first level L1. NMI-2 is the NMI between the identified partition and the predefined partition at the second level L2.

the original *Significance*, could only identify these heterogeneous communities predefined at the micro level. However, their corresponding multi-resolution versions may well detect all these predefined communities at both the macro and micro levels by adjusting the resolution parameter, as marked by L1 and L2 in Fig 9. In addition, with the increase of $\mu_1$, the resolution parameter $\gamma$ is required to reach a larger value for detecting the predefined communities at the macro level. Obviously, a large $\mu_1$ means the relatively dense links between two different macro-level communities which naturally causes these macro communities would be not easily split. Thus, a relative large value of $\gamma$ is needed to identify the macro-level communities.

In general, both the homogeneous and heterogeneous hierarchical networks include only two hierarchical levels of communities. Naturally, one can expect that these multi-resolution methods can be competent for community identification in the networks with several levels or scales of organization. In view of the hierarchical organization of many real-world networks, Yang et al. proposed a good hierarchical benchmark graph for testing various community detection algorithms [64]. The hierarchical benchmark is constructed by combining the LFR

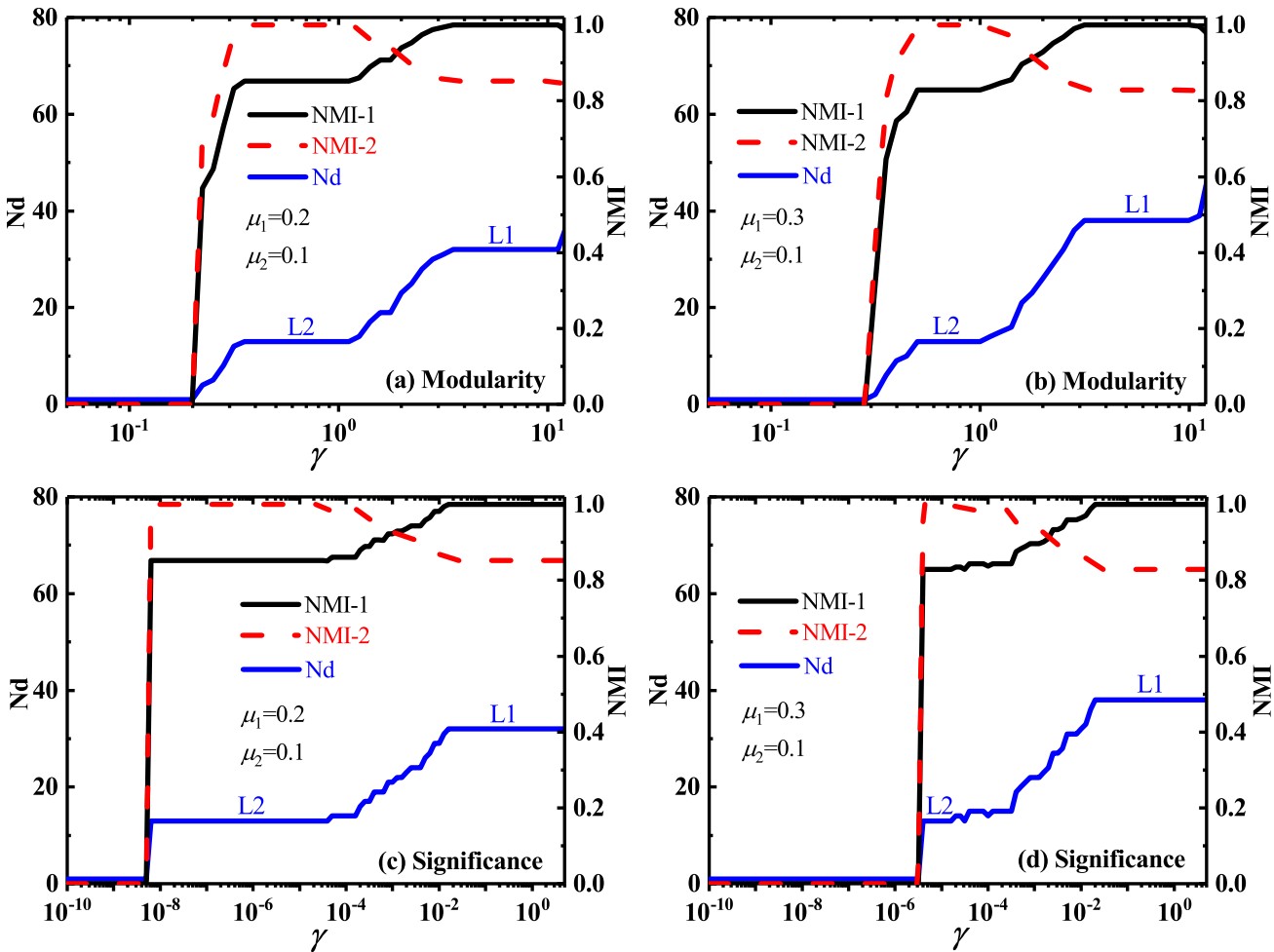

**Fig 9. Number of identified communities and normalized mutual information in heterogeneous hierarchical networks.** Number of identified communities Nd and Normalized mutual information NMI, by the multi-resolution *Modularity* (a)-(b) and the multi-resolution *Significance* (c)-(d), versus the resolution parameter $\gamma$. The heterogeneous hierarchical networks is composed of two hierarchical levels of heterogeneous communities. L1 and L2 are marked to highlight the predefined scales at the micro and macro levels, respectively. NMI-1 denotes the NMI between the identified partition and the predefined partition at the micro level L1. NMI-2 is the NMI between the identified partition and the predefined partition at the macro level L2.

benchmark graphs and the rule of constructing hierarchical organization proposed by Ravasz and Barabási, and thus is named as the Ravasz-Barabási-Lancichinetti-Fortunato-Radicchi (RB-LFR) benchmark [64]. Besides the properties of the standard LFR network, the RB-LFR benchmark possess a clear hierarchical organization with an arbitrary number of levels, which is a challenging benchmark for various community-detection methods. In the present paper, we employed the RB-LFR networks with three levels to test the multi-resolution versions of *Modularity* and *Significance*. The results have been shown in Fig 10. For two typical mixing parameters of seed LFR benchmark, three different ground truths: seed-replica-replica (abbreviated to S-R*2), replica-replica-seed (abbreviated to R*2-S) and Flat, are well identified.

As should be noted in Fig 10 that for a single mixing parameter, only two ground truths can be well defined. In order to obtain a richer hierarchical community structure, we thus extended the RB-LFR network by setting different probabilities of randomly removing connections between the seed communities and the replicas for the different hierarchies. In these extended RB-LFR benchmarks with three levels, three different community structures

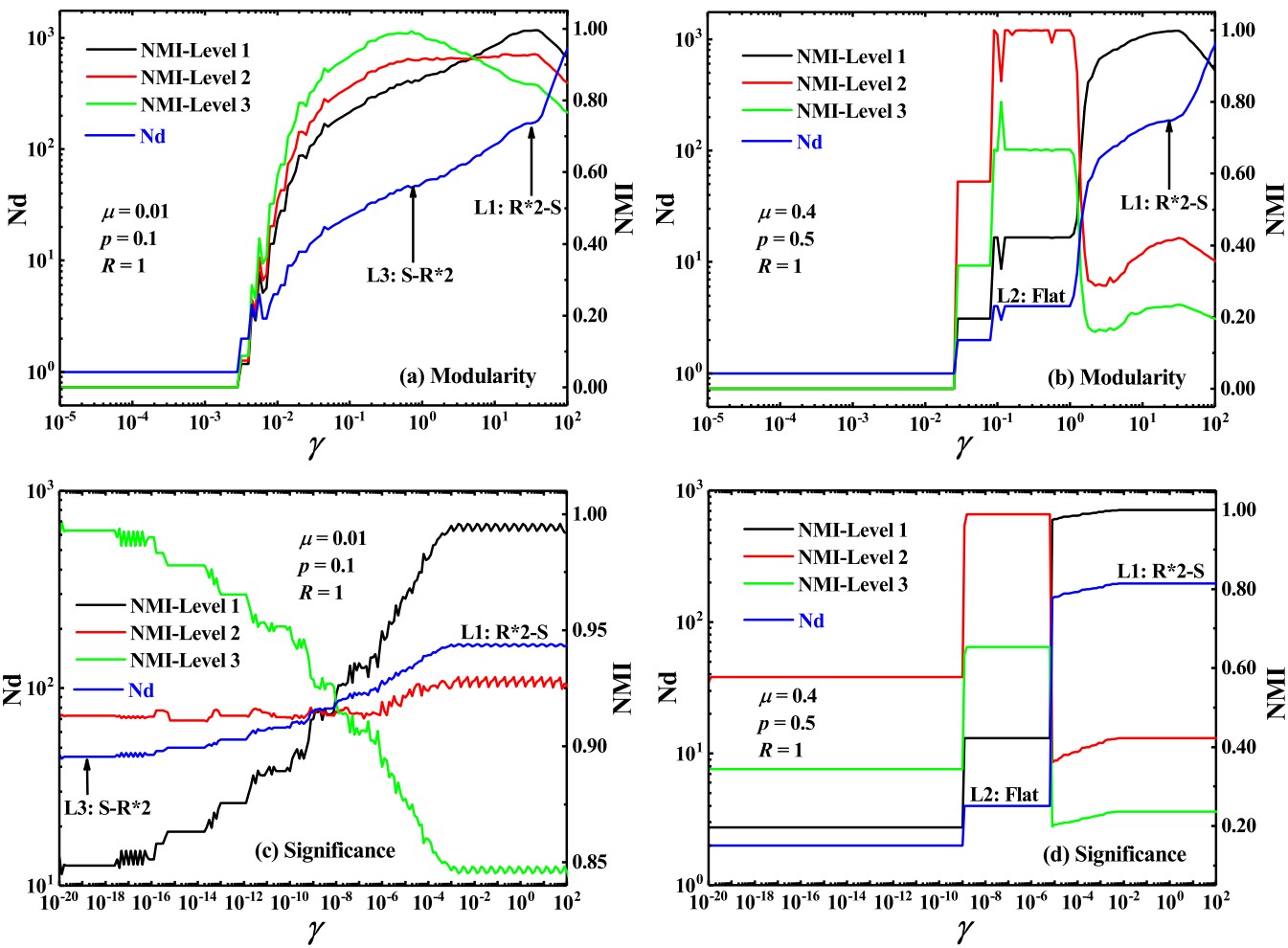

**Fig 10. Number of identified communities and normalized mutual information in the RB-LFR networks.** Number of identified communities Nd and Normalized mutual information NMI, by the multi-resolution *Modularity* and the multi-resolution *Significance* (top and bottom, respectively), as a function of resolution parameter $\gamma$, in the RB-LFR networks with three hierarchical levels. The mixing parameters are $\mu = 0.01$ and 0.4 for the left and right panels, respectively. L1, L2, and L3 denote the three predefined levels in the networks. "S-R*2" is an abbreviation of "seed-replica-replica", and "R*2-S" is an abbreviation of "replica-replica-seed". Thus, "L1: S-R*2" denote the ground truth of seed-replica-replica at level 1, while "L3: R*2-S" denote the ground truth of replica-replica-seed at level 3.

corresponding to three different hierarchies may be well defined for each of the mixing parameters. For instance, when the mixing parameter is small enough (e.g., $\mu = 0.01$) and the probabilities $p_1$ and $p_2$ of removing connections are small (e.g., $p_1 = 0.1$ and $p_2 = 0.3$), the communities for every LFR (including seed LFR and its replicas) can been well defined on the first level (or upper level), and two levels of community structures (i.e., two seed-replica-replicas), corresponding to the second and the third hierarchy, can be then defined. When the mixing parameter is large (e.g., $\mu = 0.4$) and the probabilities $p_1$ and $p_2$ of removing connections are large enough (e.g., $p_1 = 0.5$ and $p_2 = 0.9$), the first level is the same as the case for small mixing parameter, and the second and third levels are refereed to two kinds of Flats. The testing results of the multi-resolution *Significance* and *Modularity* in the extended RB-LFR networks are shown in Fig 11. It is found that when the mixing parameter is small, it is very difficult for the multi-resolution *Modularity* to detect three different community structures corresponding to three levels of organization, while the multi-resolution *Significance* can still plausibly

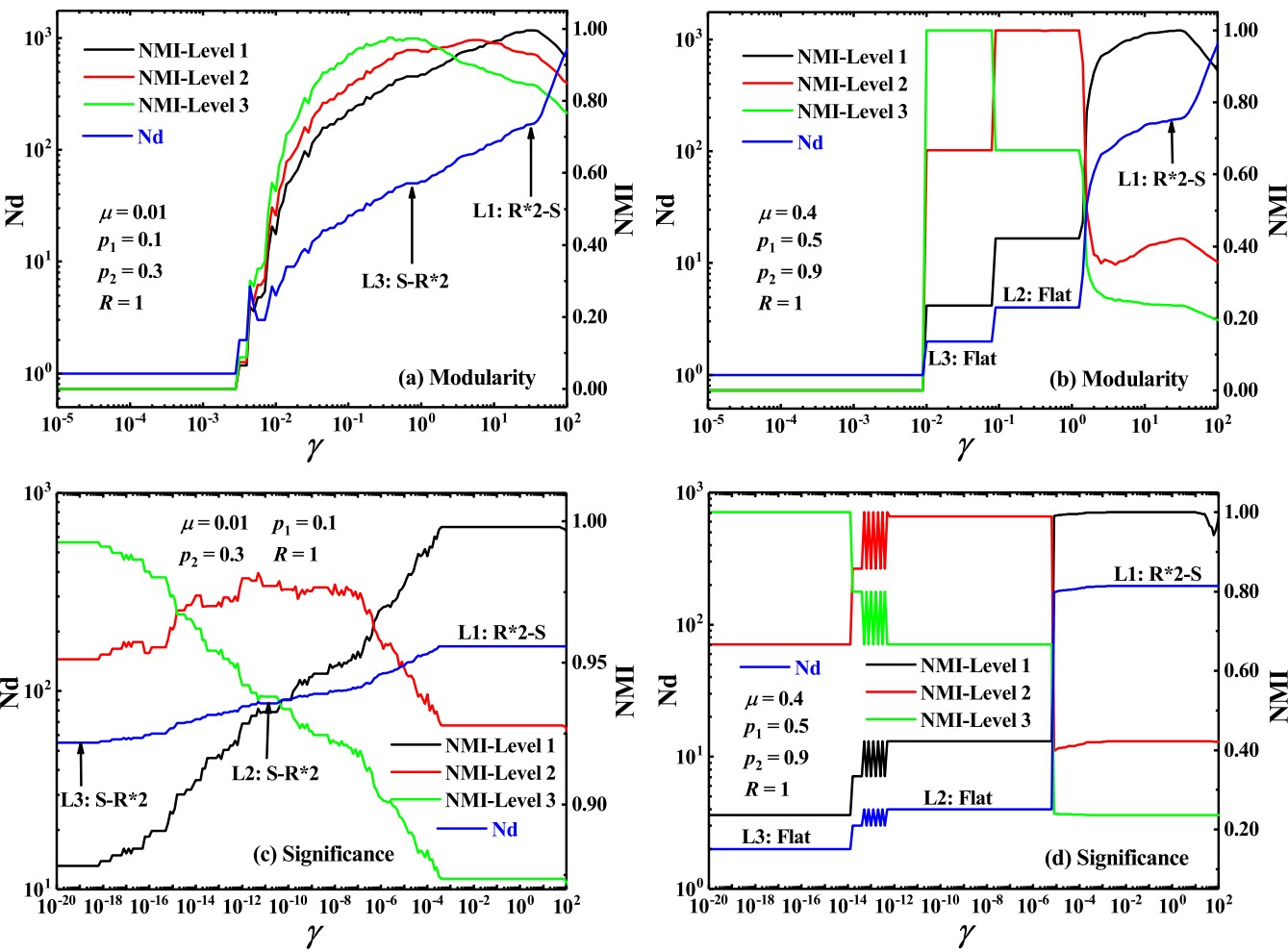

**Fig 11. Number of identified communities and normalized mutual information in the extended RB-LFR networks.** Number of identified communities Nd and Normalized mutual information NMI, by the multi-resolution *Modularity* and the multi-resolution *Significance* (top and bottom, respectively), as a function of resolution parameter γ, in the RB-LFR networks with three hierarchical levels. The mixing parameters are μ = 0.01 and 0.4 for the left and right panels, respectively. The levels and the ground truths at various levels are denoted as the corresponding abbreviations with the same rule of Fig 10.

identify all ground truths at three levels due to its high resolution (see Fig 11(a) and 11(c)). For the case of large mixing parameter, both the multi-resolution *Modularity* and *Significance* can successfully identify the predefined community structures at every level (see Fig 11(b) and 11(d)). However, it should be noted that the multi-resolution *Significance* can detect the ground truth of replica-replica-seed at level 1 (i.e., L1: R*2-S) more explicitly than the multi-resolution *Modularity* because *Significance* seems to be more competent for the detection of small-scale communities than other methods such as *Modularity*.

## 4 Application to the disease-gene identification

Module/community structure is ubiquitous in biomedical networks. Network module analysis is an important method for biomedical network research [65]. Here, we apply the multi-scale significance method to some hot issues in current computational biology: the disease-gene identification [66]. Identifying disease-related genes is of interest in the study of molecular

mechanism of disease. It is of great significance for the diagnosis, treatment and prognosis analysis of diseases.

Many candidate gene prioritization methods have been proposed based on protein network analyses. The theoretical basis of the network-based methods is that genes associated with the same or similar disease phenotypes are not randomly distributed in the network, and tend to form disease-gene modules together [67, 68]. Module structure is an important property of protein networks [69]. It is clear that protein functions arise from modular characteristics and that mutations of proteins in the same module can lead to similar disease phenotypes [70]. We have showed that identifying (single-scale) disease-related modules/communities is helpful to the identification of disease-related genes [71]. Here, we further extract multiple-scale module partitions from an integrated protein network [72] by using our multi-scale significance and then score all modules and genes according to the fraction of disease-related genes in modules so as to identify unknown disease-related genes (denoted by MSS).

Network-based methods for identifying disease-related genes at least need two types of data: protein networks and known disease-related genes for a disease. Here, we used the integrated protein network that consists of physical protein interactions from several sources [72]: literature-curated datasets, regulatory interactions, binary interactions from several yeast two-hybrid high-throughput, metabolic enzyme-coupled interactions, protein complexes, kinase-substrate pairs and signaling interactions. The set of diseases was manually chosen by a medical expert with the additional criteria of at least 20 associated genes reported in the literature [72, 73]. The associations between genes and diseases were retrieved from OMIM (Online Mendelian Inheritance in Man) and GWAS (Genome-Wide Association Studies).

We perform a 5-fold cross-validation for each disease. The disease gene set of each disease was split into five parts, one of which is used as a test set and the rest as a training set for scoring all modules and genes. Here, we calculate the proportion of disease genes in each module directly as the score of genes in the module to evaluate the probability that these genes are disease-related genes. The results show that this approach could have considerable predictive performance for identifying disease genes. For example, our module-based prioritization method outperforms classical RWR [74] for the diseases: Cardiomyopathies, Hypertrophic Cardiomyopathy, Coronary Artery Disease, Muscular Dystrophies, Mycobacterium Infections, Hereditary Spastic Paraplegia and Varicose Veins (see Fig 12(a)), and it outperforms both RWR and PRINCE [75] for the diseases: Coronary Artery Disease, Muscular Dystrophies, Mycobacterium Infections, Hereditary Spastic Paraplegia, Varicose Veins (see Fig 12(b)).

As we note, MSS uses totally different ways from other methods (e.g., RWR and PRINCE) to extract disease gene information, and the complementation for different information may bring about better results. Therefore, we try to accumulate the two scores of MSS and RWR linearly to get a comprehensive gene score (denoted by MSS+). As expected, MSS+ effectively improved the prediction performance of RWR and MSS(see Fig 12(c)). To display the improvements more clearly, we further calculate the increment of AUC for each disease obtained by MSS+ compared to MSS, RWR and PRINCE (see Fig 12(d)). It is clear that MSS+ significantly improved the performance for predicting disease-related genes for most diseases.

## 5 Conclusion

Identification of community structures has been a subject of interest in network science due to its important implications for understanding the structures and functions of complex networks. Many methods have been proposed, aiming to find a optimal network partition. Particularly, some statistical measures, such as *Modularity*, *Hamiltonian*, *Surprise* and *Significance*,

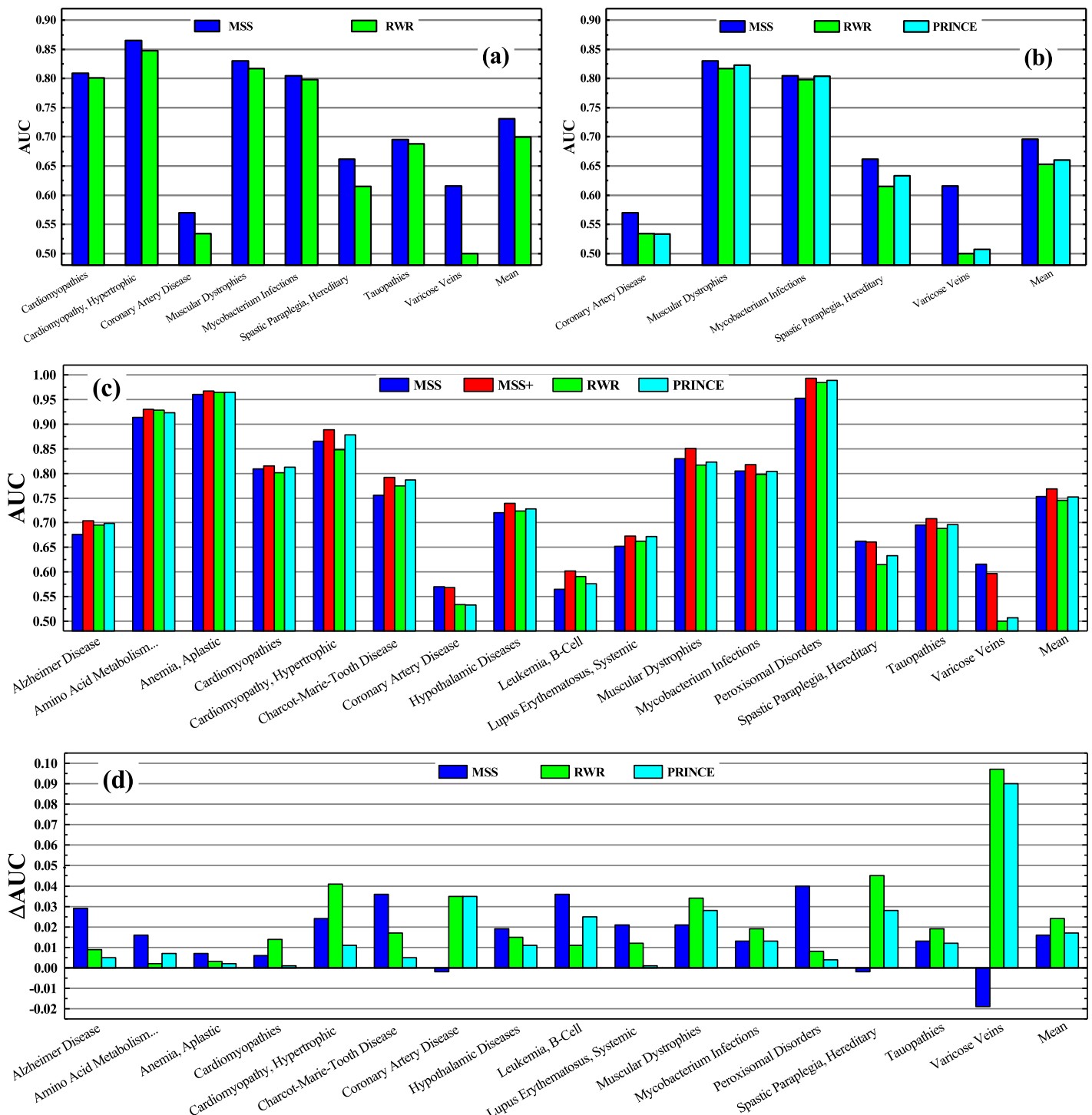

**Fig 12. Comparison of four methods of disease-gene prediction.** Comparison of Area Under ROC (receiver operating characteristic) Curves (AUCs) for several diseases: (a) MSS vs RWR and (b) MSS vs PRINCE. (c) Comparison of AUCs obtained by MSS, MSS+, RWR and PRINCE. (d) Improvements of AUCs (ΔAUC) obtained by MSS+, compared to MSS, RWR and PRINCE.

have been widely used for community detection, and also provided a basic framework for developing new methods. However, it has been recognized that all of these methods have their respective application scopes, and thus it is necessary to learn about the critical behaviors of these traditional measures. Also, some of these methods lack the flexibility of resolution, leading to the resolution limit, and thus are not compatible with the multi-scale community structures of complex networks.

In this paper, we have investigated a statistical measure of interest for community detection, i.e., *Significance*. The critical behaviors of *Significance* in community detection were analytically studied based on the derived critical number of communities where the phase diagrams for three distinct methods (*Modularity*, *Surprise* and *Significance*) were presented to demonstrate the phase transitions in network partition and compare their resolutions. It has been revealed that *Significance* generally has far higher resolution than *Modularity*. Of course, the resolutions is closely related to the intra- and inter-link densities of communities. Moreover, the so-called "potential well" effect will be encountered for *Modularity*, while not for *Significance*.

Thanks to the critical analysis, we have developed a multi-resolution *Significance*, which is a generalization of *Significance* to detect the multi-scale communities in complex networks. In several computer-generated networks, the multi-resolution *Significance* has been tested and also compared with the multi-resolution *Modularity*. The results show that, similarly to the multi-resolution *Modularity*, the multi-resolution *Significance* can well solve the first-type resolution limit by adjusting its resolution parameter. In addition, it has been demonstrated that both the multi-resolution *Significance* and *Modularity* would suffer from the second-type resolution limit. Interestingly, the multi-resolution *Significance* has stronger tolerance against the second-type resolution limit than the multi-resolution *Modularity*, due to its high resolution. In several hierarchical networks with two or three levels, we have examined the performance of the multi-resolution *Significance* and find that it can well detect the multi-scale communities in these networks.

Finally, the multi-scale *Significance* has been applied to the disease-gene prediction. The results show that extracting information from the perspective of multi-scale module mining is helpful for disease gene prediction, and its combination with other methods can effectively improve the overall performance of prediction methods. However, there are still many issues worthy of further in-depth study, e.g., how to extract the multi-scale module partitions of the networks more effectively and how to combine information from different module partitions effectively and so on. We will further study these issues in the next work.

In summary, we presented a detail critical analysis of *Significance* for community detection, and proposed an alternative *Significance*-based approach to detect the multi-scale community structures in complex networks. The results could be helpful for further understanding the behavior of *Significance*, and provide useful insight into the investigation of community structure in complex networks. Also, it has an important implication to develop any other extension of *Significance* in the future.

## Appendix A

### Critical condition of communities in partition transition

Consider a community-loop network consisting of $r$ communities with $n_c$ vertices. For the pre-defined community partition, $p_1 = \frac{n_c^2 p_i}{n_c^2} = p_i$, and $p = \frac{r \cdot n_c^2 p_i + 2r \cdot n_c^2 p_o}{r^2 n_c^2} = \frac{p_i + 2p_o}{r}$. So,

$$
\begin{aligned}
S_1 &= \frac{1}{2} r n_c (n_c - 1) D(p_1 \parallel p) \\
&= \frac{1}{2} r n_c (n_c - 1) \left( p_1 \ln \frac{p_1}{p} + (1 - p_1) \ln \frac{1 - p_1}{1 - p} \right),
\end{aligned}
$$

For the partition with $r/2$ groups of predefined communities, each of which has $2n_c$ vertices, $p_2 = \frac{2n_c^2 p_i + 2n_c^2 p_o}{4n_c^4} = \frac{p_i + p_o}{2}$. Thus,

$$
\begin{aligned}
S_2 &= \frac{1}{2} r n_c (2n_c - 1) D(p_2 \parallel p) \\
&= \frac{1}{2} r n_c (2n_c - 1) \left( p_2 \ln \frac{p_2}{p} + (1 - p_2) \ln \frac{1 - p_2}{1 - p} \right) \\
&\approx r n_c^2 \left( p_2 \ln \frac{p_2}{p} + (1 - p_2) \ln \frac{1 - p_2}{1 - p} \right).
\end{aligned}
$$

If $S_2 - S_1 > 0$, the partition with $r/2$ groups of predefined communities will be preferred, that is to say,

$$
\binom{2n_c}{2} D(p_2 \parallel p) - 2 \binom{n_c}{2} D(p_1 \parallel p) > 0. \qquad \text{(A1)} \tag{A1}
$$

By solving this equation for $r$, one can obtain the critical number of communities for *Significance*.

## Appendix B

### Analysis of "potential well" effect in community detection

Generally, statistical measures for community detection, e.g., Modularity, allow one (or two) group(s) of $x$-communities merging when a partition with $r/x$ groups of $x$-communities is allowed, or say, one (or two) group(s) of $x$-communities merging can lead to the increase of the statistical measures. However, a "potential well" effect may occur due to the nonlinearity of some statistical measures—one (or two) group(s) of $x$-communities merging can lead to the decrease of the statistical measures even if a partition with $r/x$ groups of $x$-communities has higher values of statistical measures. Surprise and its asymptotical approximation have been conformed to have the "potential well" effect, which may lead that general greedy divisive algorithms may be unable to search for global optimum effectively.

Fortunately, Significance doesn't have this effect. Consider the partition with $k$ groups of 2 predefined communities merging,

$$
\begin{aligned}
S_2(k) &= (r - 2k) \binom{n_c}{2} D(p_1 \parallel p) + k \binom{2n_c}{2} D(p_2 \parallel p) \\
&= k \left\{ k \binom{2n_c}{2} D(p_2 \parallel p) - 2 \binom{n_c}{2} D(p_1 \parallel p) \right\} + r \binom{n_c}{2} D(p_1 \parallel p).
\end{aligned}
$$

This means that $S_2(k)$ is a monotonically increasing function with the number $k$ of groups of 2 predefined communities merging. One (or two) group(s) of $x$-communities merging can be allowed as long as $\binom{2n_c}{2} D(p_2 \parallel p) - 2 \binom{n_c}{2} D(p_1 \parallel p) > 0$, that is, Eq A1 is satisfied, because $S_2(k)$ can increase with communities merging. So *Significance* does not show the "potential-well" effect.

## Acknowledgments

We are grateful to all participants for the project of National Natural Science Foundation of China and the investigators at Application Characteristic Discipline of Hunan Province.

## Author Contributions

**Conceptualization:** Ke Hu, Ju Xiang.

**Data curation:** Ke Hu, Yun-Xia Yu, Yan Zhang.

**Formal analysis:** Yun-Xia Yu, Liang Tang, Qin Xiang, Jian-Ming Li, Yong-Hong Tang, Yong-Jun Chen.

**Funding acquisition:** Ju Xiang, Jian-Ming Li.

**Investigation:** Ke Hu, Ju Xiang, Liang Tang, Qin Xiang.

**Methodology:** Ke Hu, Ju Xiang, Jian-Ming Li, Yong-Hong Tang, Yong-Jun Chen.

**Validation:** Jian-Ming Li, Yan Zhang.

**Visualization:** Yun-Xia Yu, Liang Tang, Qin Xiang.

**Writing – original draft:** Ke Hu, Ju Xiang, Yun-Xia Yu.

**Writing – review & editing:** Ke Hu, Ju Xiang.

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
