## [Decision Letter · Decision Letter 0]

21 Oct 2019

PONE-D-19-25876

Multi-scale community detection in complex networks by Significance

PLOS ONE

Dear Mrs. Yu,

Thank you for submitting your manuscript to PLOS ONE. After careful consideration, we feel that it has merit but does not fully meet PLOS ONE’s publication criteria as it currently stands. Therefore, we invite you to submit a revised version of the manuscript that addresses the points raised during the review process.

We would appreciate receiving your revised manuscript by Dec 05 2019 11:59PM. To enhance the reproducibility of your results, we recommend that if applicable you deposit your laboratory protocols in protocols.io, where a protocol can be assigned its own identifier (DOI) such that it can be cited independently in the future. For instructions see: http://journals.plos.org/plosone/s/submission-guidelines#loc-laboratory-protocols

We look forward to receiving your revised manuscript.

Kind regards,

Claudio J. Tessone, PD Ph.D.

Academic Editor

PLOS ONE

**Journal Requirements:**

**Comments to the Author**

1. Is the manuscript technically sound, and do the data support the conclusions?

Reviewer #1: Yes

Reviewer #2: Yes

2. Has the statistical analysis been performed appropriately and rigorously? 

Reviewer #1: Yes

Reviewer #2: Yes

3. Have the authors made all data underlying the findings in their manuscript fully available?

Reviewer #1: Yes

Reviewer #2: No

4. Is the manuscript presented in an intelligible fashion and written in standard English?

Reviewer #1: Yes

Reviewer #2: Yes

5. Review Comments to the Author

Reviewer #1: In the manuscript "Multi-scale community detection in complex networks by Significance", the authors have investigated a statistical measure in community detection, i.e. "Significance". They have compared the resolution of significance against modularity and surprise. After that, the authors have developed a multi-resolution significance and examined the performance of this measure.

The research question of this paper is well explained and is relevant. However, I would like to make some suggestions to the paper.

(1) In "2.1 - Critical analysis of Significance", it would be great if the authors can conduct more analysis. Please see Fig. 2 ~ 4 in "Xiang, J., Li, H. J., Bu, Z., Wang, Z., Bao, M. H., Tang, L., & Li, J. M. (2018). Critical analysis of (Quasi-) Surprise for community detection in complex networks. Scientific reports, 8(1), 14459" for details.

(2) In Fig. 3, how large is the network? Does network size play a role here?

(3) In Fig. 4 and 5, the authors have compared the NMI of significance and modularity in community-loop networks and LFR networks as a function of resolution parameter. However, the x-axis of these figures have different scales. This makes it difficult to compare the results. Please fix it.

(4) Similar to (3), the scales of x-axis are different in Fig. 6 - 8. For panel (c) & (d) in these figures, could the authors increase the range of x-axis to 10^1?

(5) What is the computational complexity of multi-resolution significance?

Overall, I like the idea of this paper and I hope my comments help in the development of the paper.

Reviewer #2: The work by K. Hu et al. focuses on the problem of community detection in complex networks. In particular, it comparatively studies the "Significance" of Traag et al. against the more traditional "Modularity" of Girvan and Newman, for the detection of communities within networks with multi-scale structures. Moreover, the present work introduces and studies a multi-resolution variation of the "Significance", essentially encompassing the novelty of the present contribution.

In my opinion the work presents interesting results, so I recommend it for publication after the following issues are appropriately addressed.

1. The NMI may result significantly non-zero when two random partitions with large numbers of groups are compared, because random coincidences become likely in this case. Similarly, it may result in artificially large values, even when two non-random partitions are compared if these have a large number of groups. To counter balance for such bias, several metrics alternative to the NMI were introduced (see [E1-E3]). It seems that Significance tends to favor the detection of small-scale structures, potentially returning partitions with more communities (i.e. groups) than other methods such as those based on Modularity Maximization. It is convenient, then, to use one of these alternative methods to judge the benefits of the Significance as compared to that of Modularity. Otherwise, the better performance could just be the outcome of chance.

2. Significance seems particularly insensitive to the resolution parameter. In some Figures gamma runs over 14 orders of magnitude. This may become a problem with networks presenting several levels or scales of organization. See for instance [E4], where benchmark networks with more than 2 levels of hierarchical organization are introduced.

3. The description of the community-loop networks seems inadequate. Readers may find difficult to reproduce the results if they are not able to appropriately generate such networks. Please improve and clarify the description. In particular, an extra figure illustrating a few example of community-loop networks could be of help.

4. It seems there is definition of $n_s$ around Eq.~1. Please add a sentence defining $n_s$.

5. There are many grammatical and orthographic errors, and a few typos (e.g. lever instead of level). Please check and correct them. Use a check speller.

6. Please consider sharing your code, if any.

Extra references

[E1] Vinh, N. X.; Epps, J.; Bailey, J. (2009). "Information theoretic measures for clusterings comparison". Proceedings of the 26th Annual International Conference on Machine Learning - ICML '09. p. 1. doi:10.1145/1553374.1553511. ISBN 9781605585161.

[E2] Meila, M. (2007). "Comparing clusterings—an information based distance". Journal of Multivariate Analysis. 98 (5): 873–895. doi:10.1016/j.jmva.2006.11.013.

[E3] M. E. J. Newman, George T. Cantwell, Jean-Gabriel Young, Improved mutual information measure for classification and community detection (2019), https://arxiv.org/abs/1907.12581

[E4] Z. Yang, J. I. Perotti, C. J. Tessone, Hierarchical benchmark graphs for testing community detection algorithms, Phys. Rev. E 96, 052311 (2017)

6. PLOS authors have the option to publish the peer review history of their article (what does this mean?). If published, this will include your full peer review and any attached files.

Reviewer #1: No

Reviewer #2: No

---

## [Author Response · Author response to Decision Letter 0]

6 Dec 2019

Dear Dr. Claudio J. Tessone and Reviewers,

 Thank you for your comments concerning our manuscript entitled “Significance-based multi-scale method for network community detection and its application in disease-gene prediction” (ID: PONE-D-19-25876). Those comments are all valuable and very helpful for revising and improving our paper, as well as the important guiding significance to our researches. We have studied Journal requirements and reviewer’s comments carefully and have made revisions which hope meet with approval. Significant changes in the text have been marked in blue for easy check purpose. The main corrections in the paper and the responds to your comments are as flowing. 

Journal Requirements:

Please ensure that your manuscript meets PLOS ONE's style requirements, including those for file naming. The PLOS ONE style templates can be found at http://www.journals.plos.org/plosone/s/file?id=wjVg/PLOSOne_formatting_sample_main_body.pdf and http://www.journals.plos.org/plosone/s/file?id=ba62/PLOSOne_formatting_sample_title_authors_affiliations.pdf

Response: We have carefully checked the manuscript and ensure that it meets PLOS ONE’s style requirements.

Response to the reviewers’ comments:

Reviewer #1: 

General Comment: In the manuscript "Multi-scale community detection in complex networks by Significance", the authors have investigated a statistical measure in community detection, i.e. "Significance". They have compared the resolution of significance against modularity and surprise. After that, the authors have developed a multi-resolution significance and examined the performance of this measure.

The research question of this paper is well explained and is relevant. However, I would like to make some suggestions to the paper.

Response: Thanks. We are very glad that you affirm the value of our research. We are responding positively to your comments listed as follows.

Comment (1-1): (1) In "2.1 - Critical analysis of Significance", it would be great if the authors can conduct more analysis. Please see Fig. 2 ~ 4 in "Xiang, J., Li, H. J., Bu, Z., Wang, Z., Bao, M. H., Tang, L., & Li, J. M. (2018). Critical analysis of (Quasi-) Surprise for community detection in complex networks. Scientific reports, 8(1), 14459" for details.

Response: Thanks for reviewer’s useful suggestion. This reference provided a good example for critical analysis. In this submitted manuscript, for the integrity of this work, we first discussed the critical characteristics of Significance in community detection, because it is based on the critical analysis of Significance that our multi-scale method was proposed. 

In order to help readers better understand this work, we further supplemented the theoretical derivation process of the critical parameter in the phase transition, and provided the theoretical proof that there is no “potential well” effect in the significance compared with the surprise. 

In order to enrich the manuscript, we further applied our multi-scale method to a hot issue in computational biology: disease-gene identification. The results showed that extracting information from the perspective of multi-scale module mining is helpful for disease gene prediction, and its combination with other methods can effectively improve the overall performance of prediction methods (see Fig 12). This provides important insights for our next research. In the future work, we will further study the applications of the multi-scale method in computational biology. 

Thanks again for the reviewer's useful suggestion 

Comment (1-2): (2) In Fig. 3, how large is the network? Does network size play a role here?

Response: Thanks for reviewer’s useful suggestion. Indeed, the network-size effect should be considered in detail. Therefore, we comprehensively computed three metrics (NMI, AMI and ARI are suggested by the second reviewer) of these methods for the different network sizes. In Fig. 4, the results for three metrics have been added. In the LFR networks, all metrics indicate that Significance gets the somewhat better performance than Surprise, and significantly overcomes Modularity. Also, six subfigures have been added to demonstrate the network-size effects (see Fig. 4(d-i) in text). Interestingly, with the increase of network size, NMI, AMI and ARI for both Significance and Surprise gradually increase, while decrease for Modularity, indicating that Significance and Surprise have better performance for the large networks than Modularity.

Comment (1-3): (3) In Fig. 4 and 5, the authors have compared the NMI of significance and modularity in community-loop networks and LFR networks as a function of resolution parameter. However, the x-axis of these figures have different scales. This makes it difficult to compare the results. Please fix it.

Authors’ Response: Thanks you for pointing out this question. For ease to compare, in the revised manuscript, we adjusted the scales in these Figures. In addition, the main purpose of Fig.4 and 5 is to demonstrate the region of resolution parameter where the predefined communities can be exactly identified. It was found: (1) when gamma<1, Significance has successfully identified all predefined communities, while for Modularity, in order to exactly detect all predefined communities, it is required that the resolution parameter must be larger than 1. (2) In order to exactly indentify all predefined communities, the region of resolution parameter for Significance is wider than that for Modularity, indicating that our method can find out the predefined community structure more easily than the multi-resolution Modularity. 

Comment (1-4): (4) Similar to (3), the scales of x-axis are different in Fig. 6 - 8. For panel (c) & (d) in these figures, could the authors increase the range of x-axis to 10^1?

Authors’ Response: As has been demonstrated in Fig.4 and 5, Significance and Modularity show the different regions of resolution parameter where the predefined network partition can be exactly detected. Thus, in order to demonstrate those promising network partitions and the tolerance to the second-type resolution limit, the scales of resolution parameters are set to be different for different methods. 

Because Significance has a higher resolution, in the region of gamma<1, it has successfully detected the predefined community structure(Figs. 7-8). Therefore, the results for gamma>1 did not continue to be shown. 

Comment (1-5): (5) What is the computational complexity of multi-resolution significance?

Authors’ Response: This is an important problem. In general, the multi-resolution Significance divided the networks into communities at each resolution by Louvain process. Louvain process is a widely used and efficient algorithm, but its exact computational complexity is not known1. Most of its computational effort is spent on the optimization at the first level, taking a time O(nkmf) if we control the maximal iteration times, where n is the number of nodes, km is the mean degree of nodes, and f is the number of operations of calculating S-value each time (on average the number of communities that each node connects to is less than the number of neighbors of the vertex). 

1https://perso.uclouvain.be/vincent.blondel/research/louvain.html

Comment (1-6): Overall, I like the idea of this paper and I hope my comments help in the development of the paper.

Authors’ Response: Thanks you for your valuable comments. All of them are very helpful for improving our manuscript. We appreciate for your warm work earnestly, and hope that these corrections will meet with your approval.

Reviewer #2: 

The work by K. Hu et al. focuses on the problem of community detection in complex networks. In particular, it comparatively studies the "Significance" of Traag et al. against the more traditional "Modularity" of Girvan and Newman, for the detection of communities within networks with multi-scale structures. Moreover, the present work introduces and studies a multi-resolution variation of the "Significance", essentially encompassing the novelty of the present contribution.

In my opinion the work presents interesting results, so I recommend it for publication after the following issues are appropriately addressed.

Response: Thanks. We are very glad that you affirm the value of our research. We are responding positively to your questions pointed out in the referee report.

Comment (2-1): 1. The NMI may result significantly non-zero when two random partitions with large numbers of groups are compared, because random coincidences become likely in this case. Similarly, it may result in artificially large values, even when two non-random partitions are compared if these have a large number of groups. To counter balance for such bias, several metrics alternative to the NMI were introduced (see [E1-E3]). It seems that Significance tends to favor the detection of small-scale structures, potentially returning partitions with more communities (i.e. groups) than other methods such as those based on Modularity Maximization. It is convenient, then, to use one of these alternative methods to judge the benefits of the Significance as compared to that of Modularity. Otherwise, the better performance could just be the outcome of chance.

Authors’ Response: Thanks for the reviewer’s useful suggestion. For comparison, we added the results of the adjusted mutual information (AMI) and adjusted Rand index (ARI). We find that the results for both ARI and AMI are similar to those of NMI, which indicate the better performance of Significance (see Fig 4 a, b, and c). Also, in Fig 4, we present the results of network-size effect to demonstrate the performances of these methods.

Comment (2-2): 2. Significance seems particularly insensitive to the resolution parameter. In some Figures gamma runs over 14 orders of magnitude. This may become a problem with networks presenting several levels or scales of organization. See for instance [E4], where benchmark networks with more than 2 levels of hierarchical organization are introduced.

Authors’ Response: In reference E4, Yang et al. proposed a good hierarchical benchmark graph (named as RB-LFR network) for testing various community detection algorithms. In our manuscript, we employed the RB-LFR networks with three levels to test both Modularity and Significance. The results have been shown in Fig 10. For two typical mixing parameters of seed LFR benchmark, three different ground truths: seed-replica-replica, replica-replica-seed and flat, are well identified. In order to obtain a richer hierarchical community structure, we also extended the RB-LFR network by setting different probabilities of randomly removing connections between the seed communities and the replicas for the different hierarchies. In these extended RB-LFR benchmarks with three levels, three different community structures corresponding to three different hierarchies can be well defined for each of mixing parameters. For instance, when the mixing parameter is small enough (e.g., μ=0.01) and the probabilities p1 and p2 of removing connections are small (e.g., p1=0.1 and p2=0.3), the communities for every LFR (including seed LFR and its replicas) can been well defined on the first level (or upper level), and two levels of community structures (i.e., two seed-replica-replicas), corresponding to the second and the third hierarchy, can be then defined. When the mixing parameter is large (e.g., μ=0.4) and the probabilities p1 and p2 of removing connections are large enough (e.g., p1=0.5 and p2=0.9), the first level is the same as the case for small mixing parameter, and the second and third levels are refereed to two kinds of flats. In view of the explicit hierarchical community structures of the extended RB-LFR benchmark, we also test both Significance and Modularity in these benchmark networks. The results show that the multi-resolution Significance and Modularity can well identify the predefined community structures at every level (see Fig 11).

Thanks you for your valuable suggestion.

Comment (2-3): 3. The description of the community-loop networks seems inadequate. Readers may find difficult to reproduce the results if they are not able to appropriately generate such networks. Please improve and clarify the description. In particular, an extra figure illustrating a few example of community-loop networks could be of help.

Authors’ Response: We added a figure (i.e., Fig 1) illustrating an example of community-loop networks. Thanks for your useful suggestion.

Comment (2-4): 4. It seems there is definition of $n_s$ around Eq.~1. Please add a sentence defining $n_s$.

Authors’ Response: Thanks you for pointing out this question. We added a sentence defining $n_s$.

Comment (2-5): 5. There are many grammatical and orthographic errors, and a few typos (e.g. lever instead of level). Please check and correct them. Use a check speller.

Authors’ Response: We revised and checked the manuscript carefully. Thanks you for these useful suggestions.

Comment (2-6): 6. Please consider sharing your code, if any.

Authors’ Response: The readers could request the code of the paper by the corresponding author (Email: yu.sunny@126.com). 

Moreover, the Louvain algorithm has public codes in R, Python and C++:

https://www.rdocumentation.org/packages/igraph/versions/1.2.4/topics/cluster_louvain

https://pypi.org/project/louvain/

https://perso.uclouvain.be/vincent.blondel/research/louvain.html

Extra references

[E1] Vinh, N. X.; Epps, J.; Bailey, J. (2009). "Information theoretic measures for clusterings comparison". Proceedings of the 26th Annual International Conference on Machine Learning - ICML '09. p. 1. doi:10.1145/1553374.1553511. ISBN 9781605585161.

[E2] Meila, M. (2007). "Comparing clusterings—an information based distance". Journal of Multivariate Analysis. 98 (5): 873–895. doi:10.1016/j.jmva.2006.11.013.

[E3] M. E. J. Newman, George T. Cantwell, Jean-Gabriel Young, Improved mutual information measure for classification and community detection (2019), https://arxiv.org/abs/1907.12581

[E4] Z. Yang, J. I. Perotti, C. J. Tessone, Hierarchical benchmark graphs for testing community detection algorithms, Phys. Rev. E 96, 052311 (2017)

We tried our best to revise our manuscript according to the comments. Also, several minor problems have been corrected. Four figures (Fig 1, 10, 11, and 12) and some references (including E1-E4) have been added, and thus the figures and references are renumbered. Fig 4 has been recalculated and re-plotted.

Once again, we would like to express our great appreciation to you for helpful comments on our manuscript.

With our thanks and best regards.

Yours sincerely,

Yun-Xia Yu

E-mail: yu.sunny@126.com

---

## [Editor Report · Decision Letter 1]

17 Dec 2019

Significance-based multi-scale method for network community detection and its application in disease-gene prediction

PONE-D-19-25876R1

Dear Dr. Yu,

We are pleased to inform you that your manuscript has been judged scientifically suitable for publication and will be formally accepted for publication once it complies with all outstanding technical requirements.

With kind regards,

Claudio J. Tessone, PD Ph.D.

Academic Editor

PLOS ONE

Additional Editor Comments (optional):

Dear Dr Yun-Xia Yu

We are happy to confirm that your manuscript entitled "Significance-based multi-scale method for network community detection and its application in disease-gene prediction" has been accepted for Publication in PLOS ONE. This decision follows from your careful reply to the reviewer's comments.
---

## [Editor Report · Acceptance letter]

4 Mar 2020

PONE-D-19-25876R1 

Significance-based multi-scale method for network community detection and its application in disease-gene prediction 

Dear Dr. Yu:

I am pleased to inform you that your manuscript has been deemed suitable for publication in PLOS ONE. Congratulations! Your manuscript is now with our production department. 

With kind regards,

on behalf of

Dr. Claudio J. Tessone 

Academic Editor

PLOS ONE